# Neuroprotective Effects of Geopung-Chunghyuldan Based on Its Salvianolic Acid B Content Using an In Vivo Stroke Model

Han-Gyul Lee [1], Seungwon Kwon [1,*], Sang-Kwan Moon [1,*], Seung-Yeon Cho [2], Seong-Uk Park [2], Woo-Sang Jung [1], Jung-Mi Park [2], Chang-Nam Ko [2] and Ki-Ho Cho [1]

[1] Department of Cardiology and Neurology, Kyung Hee University Medical Center, College of Korean Medicine, Kyung Hee University, Seoul 02447, Republic of Korea
[2] Department of Cardiology and Neurology, Kyung Hee University Hospital at Gangdong, College of Korean Medicine, Kyung Hee University, Seoul 02447, Republic of Korea
* Correspondence: kkokkottung@hanmail.net (S.K.); skmoon@khu.ac.kr (S.-K.M.)

**Abstract:** Background: Geopung-Chunghyuldan (GCD) has neuroprotective properties. *Salviae miltiorrhizae* Radix plays an essential role in GCD's effect. The *Salviae miltiorrhizae* Radix marker compound is salvianolic acid B; however, its content is not uniform among samples. This study aimed to evaluate the neuroprotective effects of GCD based on salvianolic acid B content. Methods: The neuroprotective effects of GCD based on the salvianolic acid B content were evaluated by measuring infarct volume 24 h after permanent middle cerebral artery occlusion in an in vivo stroke model. For the experimental group, each GCD was administered immediately before surgery. The control groups were administered distilled water and aspirin (30 mg/kg) in the same way. The salvianolic acid B content in five types of Salviae Miltiorrhizae Radix (two Chinese and three Korean regions) based on different cultivation regions was analyzed by high-performance liquid chromatography. Results: Three samples met the Korean and Chinese Pharmacopeia standards for salvianolic acid B. However, two samples did not. GCDs with high salvianolic acid B showed marked neuroprotective effects compared to the control groups, whereas GCDs with low salvianolic acid B did not. Conclusions: The salvianolic acid B content of *Salviae miltiorrhizae* Radix affects the neuroprotection effect of GCD. Stable, raw *Salviae miltiorrhizae* Radix is essential for GCD homogenization.

**Keywords:** Geopung-Chunghyuldan; *Salviae miltiorrhizae* Radix; salvianolic acid B; neuroprotection; permanent middle cerebral artery occlusion

## 1. Introduction

Stroke can be fatal or lead to irreversible neurological deficits. In the currently aging society with increased life expectancy, urgent measures are required from the perspective of medical economics as the incidence of cerebral infarction is increasing [1]. Neuroprotection has been a major target for treatment after the onset of cerebral infarction [2]. Many attempts have been made to develop neuroprotective medications after the onset of cerebral infarction; however, clinical trials have not been successful thus far [3]. Although edaravone is known to exert neuroprotective action through antioxidant effects [4], the evidence for its therapeutic effect on cerebral infarction remains insufficient [5].

Traditional East Asian Medicine (TEAM) has accumulated clinical experience in treating cerebral infarction and, thus, is attracting attention as a complementary treatment method to Western medicine that has not been completely successful to date [6–8]. Kyung Hee University Korean Medicine Hospital Stroke and Brain Diseases Center developed Geopung-Chunghyuldan (GCD) by ethanol extraction of *Salviae miltiorrhizae* Radix, *Notoginseng* Radix, *Scutellariae* Radix, *Coptidis* Rhizoma, *Phellodendri* Cortex, *Gardeniae* Fructus and *Rhei* Rhizoma, which is effective in treating and preventing cerebral infarction [9–21]. In addition, GCD was found to have a neuroprotective effect [22].

Previous studies indicate that *Salviae miltiorrhizae* Radix plays an essential role in the neuroprotective effect of GCD [23]. *Salviae miltiorrhizae* Radix (danshen), the dried root of *Salvia miltiorrhiza* BGE (Labiatae), is a very important TEAM that promotes blood flow to overcome blood stasis [24] and is currently used to treat cerebrovascular diseases, thrombosis, vasculitis, dysmenorrhea, postpartum abdominal pain, chest pain, and contusion in TEAM [25]. Previous studies suggest that *Salviae miltiorrhizae* Radix displays antiarteriosclerotic [26], anti-inflammatory [27], antidiabetic [28], antioxidant [29], antibacterial [30], anticancer [31], neuroprotective [32], immunomodulatory activity [33], antithrombotic, and antioxidant effects [34].

However, a previous study suggested that the content of salvianolic acid B, the marker compound of *Salviae miltiorrhizae* Radix, varies among the GCD products available in the market [35]. Thus, there may be a difference in the effects of GCD depending on the *Salviae miltiorrhizae* Radix content. Since the Korean Pharmacopoeia and the Chinese Pharmacopoeia set standards for marker compounds for each medication, the difference in the marker compounds content of *Salviae miltiorrhizae* Radix may seriously violate this standard [36,37]. However, no previous studies have compared the efficacy of the drug according to the marker compound content due to the difference in the materials used even with the same GCD, which is a strong justification for the content verification. Therefore, for effective and stable quality control of the neuroprotective effect of GCD, there is a need to evaluate the raw materials of *Salviae miltiorrhizae* Radix and conduct a comparative study on the neuroprotective effect of GCD, including *Salviae miltiorrhizae* Radix, with various concentrations of marker compounds.

In the current study, the salvianolic acid B content was analyzed in five different types of *Salviae miltiorrhizae* Radix. The difference in neuroprotective effects of GCD samples with varying salvianolic acid B content was compared using a permanent middle cerebral artery occlusion (pMCAO) model. Ultimately, the goal of this study was to contribute to the homogenization and standardization of GCD.

## 2. Materials and Methods

### 2.1. Experimental Drug Analyses

2.1.1. Experimental Drug Preparation

The herbal medicines used in this experiment, *Coptidis* Rhizoma, *Phellodendri* Cortex, *Scutellariae* Radix, *Gardeniae* Fructus, *Rhei* Rhizoma, *Salviae miltiorrhizae* Radix, and *Notoginseng* Radix, were purchased from Kyunghee Herbal Medicine (Wonju, Korea). Experimental drugs A and B were prepared using the following method: Herbal medicines (340 g) composed of *Coptidis* Rhizoma, *Phellodendri* Cortex, *Scutellariae* Radix, *Gardeniae* Fructus, and *Rhei* Rhizoma (4:4:4:4:1) were reflux-extracted twice with 80% ethanol in boiling water for 2 h, filtered, evaporated in a rotary vacuum evaporator, and lyophilized with a freeze dryer to produce experimental drug A. The dry weight yield was 26.5%. The basis for standardization of the preparation of experimental drug A has been suggested previously [17]. Experimental drug B (extract of *Salviae miltiorrhizae* Radix and *Notoginseng* Radix) was produced using the same method as above, using 315 g of herbal medicines composed of *Salviae miltiorrhizae* Radix and *Notoginseng* Radix 17.5:3.4, with a yield of 33.3%. The basis for standardization of the preparation of experimental drug B was suggested previously [22] (Table 1). BÜCHI Rotavapor R-220 (BÜCHI Labortechnik, Flawil, Switzerland) and deep freezer (IlShin BioBase, Dongducheon, Korea) were used to prepare the raw material extracts. For experimental drug B, five types of *Salviae miltiorrhizae* Radix (C1, C2, K1, K2, and K3) according to different cultivation regions were purchased from markets and used to prepare five different kinds of drug B (BC1 and BC2 were made using two Chinese *Salviae miltiorrhizae* Radix C1 and C2, respectively; BK1, BK2, and BK3 were prepared using three Korean *Salviae miltiorrhizae* Radix K1, K2, and K3, respectively). C1 and C2 were purchased in 2018.Feb and 2019.Jan, respectively. K1, K2, and K3 were purchased in 2019 from Yeongyang, in 2019 from Jangheung, and in 2020 from Yeongyang, respectively.

**Table 1.** Constituent herbal medicines of experimental drugs A and B.

| Experimental Drug | Herbal Medicine | Scientific Name |
|---|---|---|
| Drug A | *Coptidis* Rhizoma<br>*Phellodendri* Cortex<br>*Scutellariae* Radix<br>*Gardeniae* Fructus<br>*Rhei* Rhizoma | *Coptis japonica* MAKINO<br>*Phellodendron amurense* RUPRECHT<br>*Scutellaria baicalensis* GEORGI<br>*Gardenia jasminoides* ELLIS<br>*Rheum palmatum* LINNE |
| Drug B | *Salviae Mitiorrhizae* Radix<br>*Notoginseng* Radix | *Salvia miltiorrhiza* BUNGE<br>*Panax notoginsengs* (Burk) F. H. Chen |

2.1.2. High-Performance Liquid Chromatography (HPLC) Analysis for *Salviae mitiorrhizae* Radix

HPLC analysis was performed to analyze the content of salvianolic acid B [36] in five types of *Salviae miltiorrhizae* Radix (C1, C2, K1, K2, and K3) from different cultivation regions. The HPLC system used in this experiment was a Jasco HPLC system (LC2000PLUS, Tokyo, Japan). The column was Nucleosil C18 (5µm, 4.6 × 250 mm I.D.; MACHEREY-NAGEL, USA), used at 25 °C. Standardized salvianolic acid B was purchased from Fluka (Zwijndrecht, The Netherlands), and HPLC solvents, such as methanol, acetonitrile, and water were purchased from J.T. Baker (USA). Special-grade chemicals (Sigma, Saint Louis, MO, USA) were used as solvents for preparing other drug solutions.

To analyze salvianolic acid B, 10 mL of 85% methanol was added to 100 mg of *Salviae miltiorrhizae* Radix, followed by sonication for 30 min and filtration through a 0.45-µm polyvinylidene fluoride membrane to obtain a sample solution.

The analytical separations were conducted using a mobile phase consisting of water, acetonitrile, and acetic acid at the ratio of 750:250:10. A flow rate of 1 mL/min was employed at room temperature (25 °C). The chromatograms were monitored at a specific wavelength of 280 nm.

*2.2. Animal Experiments*

2.2.1. Animal Preparation

Seven-week-old ICR-type male mice (30 g), managed in a sterile state, were purchased from Samtaco (Seoul, Korea). Radiation-sterilized feed for experimental animals was purchased from Purina (Seoul, Korea), and feed and water were supplied freely. Experiments were started after the animals were sufficiently acclimated for 7 d in a breeding room where the temperature ($23 \pm 1$ °C) and humidity ($50 \pm 10$%) were controlled. All animal experiments were performed with the approval of the Institutional Animal Care and Use Committee (KHMC-IACUC 19-004).

2.2.2. Permanent Middle Cerebral Artery Occlusion (pMCAO) Model

Focal cerebral ischemia was induced by permanent middle cerebral artery occlusion (pMCAO) as described previously [38] with the following modifications. ICR mice weighing 30–35 g were anesthetized with 2% isoflurane in a mixture of 70% $N_2O$ and 30% $O_2$. The temporal bone was exposed through a 1.0 cm vertical temporal muscle incision midway between the left eye and ear. After checking the middle cerebral artery (MCA) with a surgical microscope, a 2.0 mm burr hole was drilled into the temporal bone, the dura was carefully removed, the main trunk was cauterized at the end of the MCA using a bipolar coagulator, and the cauterization site was cut to confirm the blockage of blood flow using a microscissor. After suturing the skin incision, a temperature maintenance device (Harvard Apparatus, Holliston, MA, USA) was used to prevent the temperature from dropping below $38 \pm 1$ °C during the recovery phase. During surgery, a heating pad was used to maintain a rectal temperature of 36.5–37.5 °C. Mice that did not turn to the paralytic side immediately or had a subarachnoid hemorrhage after surgery were excluded from the analysis.

2.2.3. Experimental Drug Administration

The experimental drugs used were a combination of drug A + 5 types of drug B. Therefore, the experimental drugs were A+BC1, A+BC2, A+BK1, A+BK2, and A+BK3, and the mixing ratio for each was the same as A:B = 60 mg/kg:30 mg/kg, which was the optimal composition identified in a previous study [23]. Control drugs other than the experimental drugs used in the experiments were as follows:

- Control: Distilled water (DW), Aspirin® (BAYER KOREA, Seoul, Korea) 30 mg/kg (ASA30).

To evaluate the neuroprotective effect of the experimental drugs, the prepared drug was administered immediately once before the pMCAO surgery by oral administration. In total, 10–15 mice were assigned to each administration group. Neuroprotective effects were evaluated by measuring infarct volume.

2.2.4. Infarct Volume Measurements

Twenty-four hours after pMCAO, mice were sacrificed by dislocation of the cervical spine and were decapitated. A rongeur was used to remove the left and right portions of the skull centering on the sulcus in the middle of the left and right hemispheres. The brains were removed and sliced into 2 mm coronal sections with a matrix (Harvard Bioscience). The brain sections were incubated in normal saline containing 1% 2, 3, 5-triphenyltetrazolium chloride (Sigma-Aldrich Korea, Seoul, Korea) at 37 °C for 20 min and subsequently stored in 10% phosphate-buffered formalin (Sigma-Aldrich Korea, Seoul, Korea). The stained and fixed brain slices were photographed using a digital camera. The total cross-sectional area and the unstained portion of each coronal slice were determined using ImageJ software (ver. 1.47). The infarct volume ($mm^3$) was calculated by measuring infarct areas on separate slices, multiplying these areas by slice thickness, and summing the areas of all slices.

2.3. *Statistical Analysis*

Data are summarized as mean ± standard deviation. Statistical analyses were performed using GraphPad Prism ver. 4 [39] or IBM SPSS Statistics ver. 25 [40]. A one-way analysis of variance (ANOVA) or Kruskal–Wallis analysis was performed for the average comparison of three or more groups and Tukey multiple comparison test was used for post-hoc analysis. Repeated measures ANOVA was used for repeatedly measured data among the groups. Statistical significance was set at $p < 0.05$.

**3. Results**

3.1. *HPLC Analysis*

In the HPLC analysis for salvianolic acid B, three types of *Salviae miltiorrhizae* Radix (K1, K2, and K3) showed a high peak value similar to that of the standardized salvianolic acid B, whereas C1 and C2 showed a low peak value (Figure 1). Salvianolic acid B content in C1 and C2 showed low values of 2.7% and 2.4%, respectively. However, salvianolic acid B content in K1, K2, and K3 showed high values of 7.4%, 6.2%, and 7.5%, respectively (Table 2).

**Table 2.** The contents of salvianolic acid B in five kinds of *Salviae Mitiorrhizae* Radix.

| Drugs | Salvianolic Acid B Content (%) |
|---|---|
| C1 (2018) | 2.7 |
| C2 (2019) | 2.4 |
| K1 (2019) | 7.4 |
| K2 (2019) | 6.2 |
| K3 (2020) | 7.5 |

C, Chinese Salviae Mitiorrhizae Radix; K, Korean Salviae Mitiorrhizae Radix.

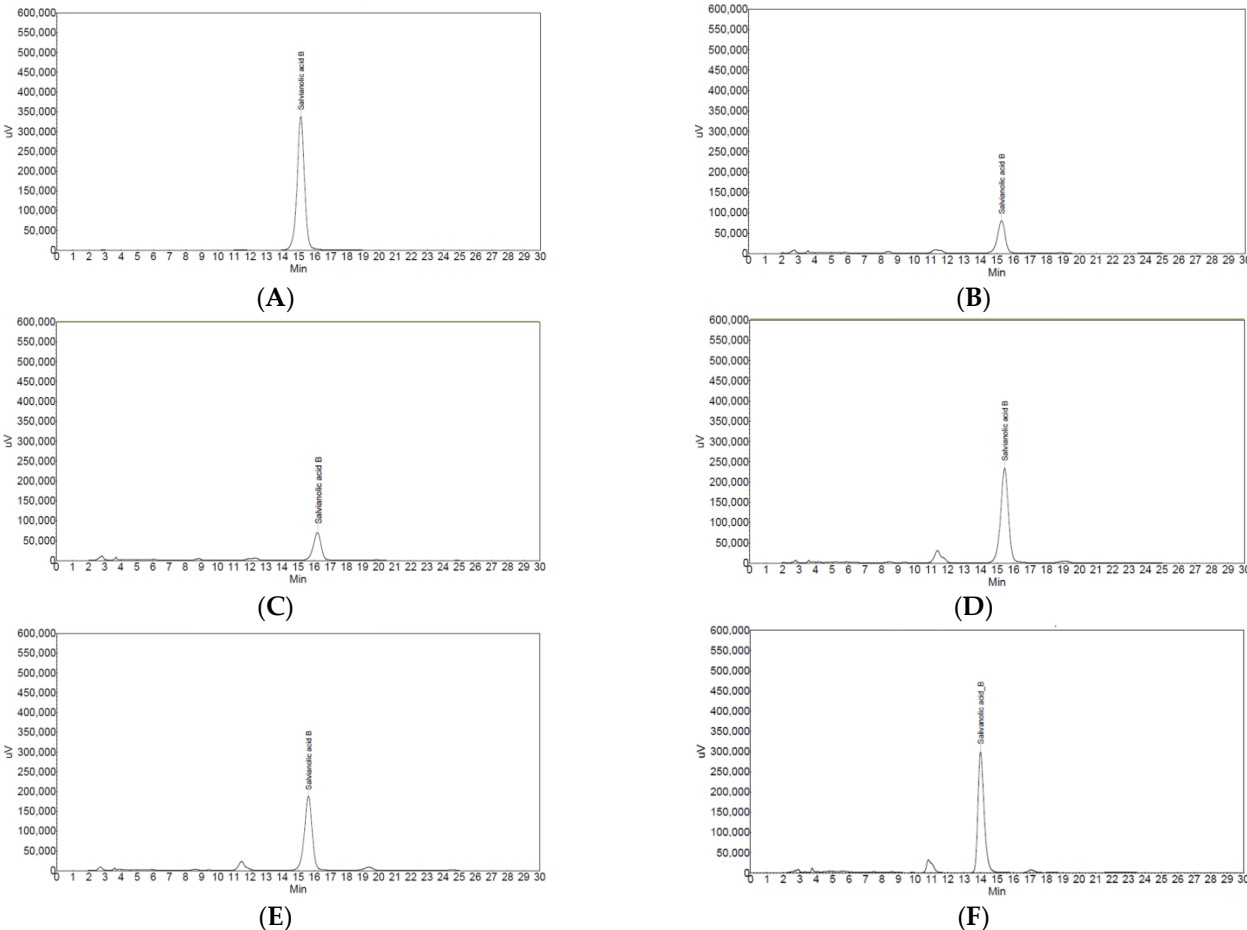

**Figure 1.** Results of high-performance liquid chromatography chromatogram for the salvianolic acid B in five types of *Salviae mitiorrhizae* Radix. (**A**) Standardized salvianolic acid B. (**B**) Salvianolic acid B in C1. (**C**) Salvianolic acid B in C2. (**D**) Salvianolic acid B in K1. (**E**) Salvianolic acid B in K2. (**F**) Salvianolic acid B in K3. K1, K2, and K3 showed high peak values similar to that of the standardized salvianolic acid B. C1 and C2 showed low peak values. C, Chinese *Salviae miltiorrhizae* Radix; K, Korean *Salviae miltiorrhizae* Radix.

### 3.2. Effects of Experimental Drugs on Infarct Volume after pMCAO

3.2.1. Effects of Experimental Drug A+BC1 on Infarct Volume after pMCAO

The experimental drug A 60 mg/kg and BC1 30 mg/kg combination (A60+B(C1)30) was administered orally before pMCAO surgery, to evaluate its neuroprotective effect. Distilled water (DW), Aspirin® 30 mg/kg (ASA30), and drug A 60 mg/kg (A60) were also administered in the same manner as controls. The infarct volume was measured 24 h after pMCAO. The experimental drug A+BC1 showed no significant effect on infarct volume after pMCAO among the groups ($p$ = 0.6408) (Table 3 and Figure 2).

**Table 3.** Effects of experimental drug A60+B(C1) 30 mg/kg on infarct volume after pMCAO.

| Variables | | Experimental Drugs (mg/kg) | | | | $p$-Value |
|---|---|---|---|---|---|---|
| | | DW (n = 17) | ASA30 (n = 15) | A60 (n = 17) | A60+B(C1)30 (n = 13) | |
| Infarct Volume | Mean (mm³) | 21.7 | 20.6 | 23.0 | 21.5 | 0.6408 |
| | SD | 4.8 | 4.9 | 5.8 | 5.6 | |

pMCAO, permanent middle cerebral artery occlusion; DW, distilled water; ASA, Aspirin®; A, Ethanol extract of *Coptidis* Rhizoma, *Phellodendri* Cortex, *Scutellariae* Radix, *Gardeniae* Fructus, and *Rhei* Rhizoma; B(C1), Ethanol extract of 2018 Chinese *Salviae Mitiorrhizae* Radix and *Notoginseng* Radix; SD, standard deviation.

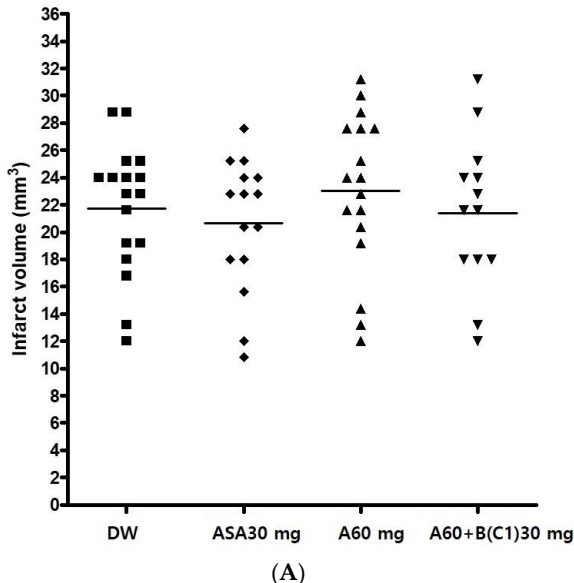

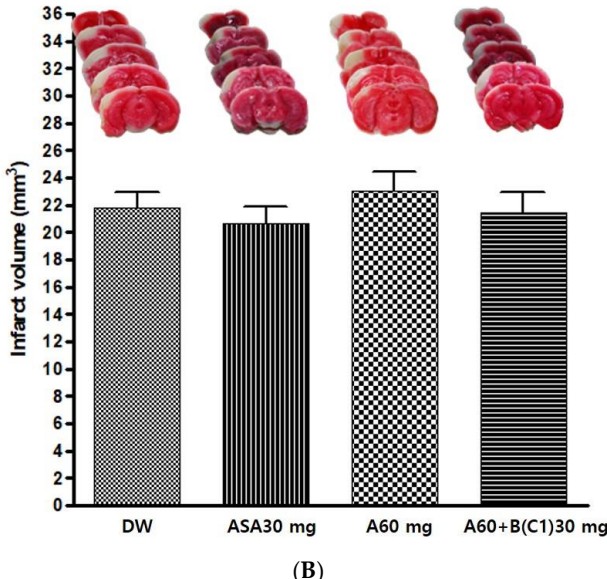

(**A**)  (**B**)

**Figure 2.** Effects of experimental drug A60+B(C1) 30 mg/kg on ischemic brain injury after permanent middle cerebral artery occlusion (pMCAO). Infarct volume was measured 24 h after pMCAO. DW and all drugs were administered just before pMCAO. (**A**) Individual infarct volume values for each group (n = 13–17 mice in each group). (**B Upper**) Representative photographs of infarcted brain slices from DW- and drug-treated mice. (**B Lower**) There was no significant difference in infarct volume among groups. DW, distilled water; ASA, Aspirin®; A, Ethanol extract of *Coptidis* Rhizoma, *Phellodendri* Cortex, *Scutellariae* Radix, *Gardeniae* Fructus, and *Rhei* Rhizoma; B(C1), Ethanol extract of 2018 Chinese *Salviae miltiorrhizae* Radix and *Notoginseng* Radix. Data represent means ± standard deviation.

### 3.2.2. Effects of Experimental Drug A+BC2 on Infarct Volume after pMCAO

The experimental drug A 60 mg/kg and BC2 30 mg/kg combination (A60+B(C2)30) was administered orally before pMCAO surgery to confirm its neuroprotective effect. DW, Aspirin® 30 mg/kg (ASA30), and drug A 60 mg/kg (A60) were also administered in the same manner as controls. The infarct volume was measured 24 h after pMCAO. The infarct volume in the group administered the experimental drug A+BC2 exhibited a decreasing tendency; however, there was no significant difference in infarct volume among groups ($p$ = 0.2989) (Table 4 and Figure 3).

**Table 4.** Effects of experimental drug A60+B(C2) 30 mg/kg on infarct volume after pMCAO.

| Variables | | Experimental Drugs (mg/kg) | | | | *p*-Value |
|---|---|---|---|---|---|---|
| | | DW (n = 17) | ASA30 (n = 15) | A60 (n = 17) | A60+B(C2)30 (n = 9) | |
| Infarct volume | Mean (mm³) | 21.7 | 20.6 | 23.0 | 19.2 | 0.2989 |
| | SD | 4.8 | 4.9 | 5.8 | 4.6 | |

pMCAO, permanent middle cerebral artery occlusion; DW, distilled water; ASA, Aspirin®; A, Ethanol extract of *Coptidis* Rhizoma, *Phellodendri* Cortex, *Scutellariae* Radix, *Gardeniae* Fructus, and *Rhei* Rhizoma; B(C2), Ethanol extract of 2019 Chinese *Salviae Mitiorrhizae* Radix and *Notoginseng* Radix; SD, standard deviation.

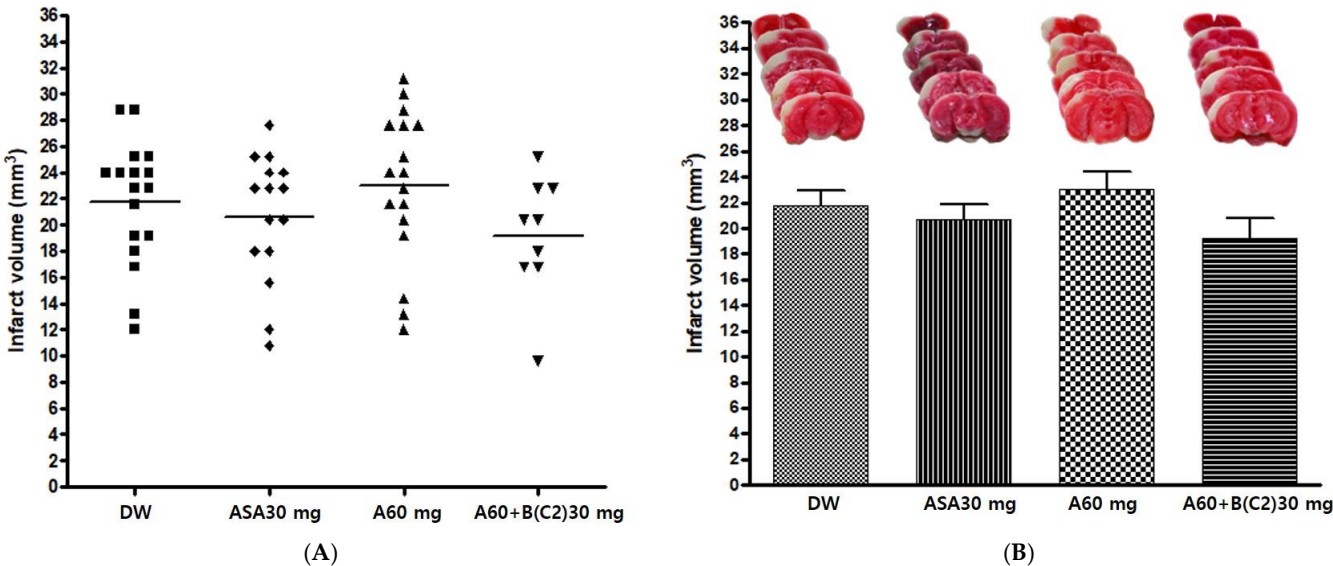

**Figure 3.** Effects of experimental drug A60+B(C2) 30 mg/kg on ischemic brain injury after permanent middle cerebral artery occlusion (pMCAO). Infarct volume was measured 24 h after pMCAO. DW and all drugs were administered just before pMCAO. (**A**) Individual infarct volume values for each group (n = 9–17 mice in each group). (**B Upper**) Representative photographs of infarcted brain slices from DW- and drug-treated mice. (**B Lower**) No significant difference in infarct volume was observed among groups. DW, distilled water; ASA, Aspirin®; A, Ethanol extract of *Coptidis* Rhizoma, *Phellodendri* Cortex, *Scutellariae* Radix, *Gardeniae* Fructus, and *Rhei* Rhizoma; B(C2), Ethanol extract of 2019 Chinese *Salviae miltiorrhizae* Radix and *Notoginseng* Radix. Data represent means ± standard deviation.

### 3.2.3. Effects of Experimental Drug A+BK1 on Infarct Volume after pMCAO

The experimental drug A 60 mg/kg and BK1 30 mg/kg combination (A60+B(K1)30) was administered orally before pMCAO surgery to confirm the neuroprotective effect. DW, Aspirin® 30 mg/kg (ASA30), and drug A 60 mg/kg (A60) were also administered in the same manner as controls. The infarct volume was measured 24 h after pMCAO. The group administered the experimental drug A+BK1 showed a significant decrease in infarct volume after pMCAO compared with other groups ($p = 0.007$). Post-hoc analysis revealed that the group administered the experimental drug A+BK1 exhibited a significant decrease in infarct volume after pMCAO compared to the groups administered DW and A60 ($p < 0.05$ and $p < 0.01$, respectively) (Table 5 and Figure 4).

**Table 5.** Effects of experimental drug A60+B(K1) 30 mg/kg on infarct volume after pMCAO.

| Variables | | Experimental Drugs (mg/kg) | | | | *p*-Value |
|---|---|---|---|---|---|---|
| | | DW (n = 17) | ASA30 (n = 15) | A60 (n = 18) | A60+B(K1)30 (n = 19) | |
| Infarct volume | Mean (mm³) | 21.7 | 20.6 | 22.3 | 17.0 *, ** | 0.007 |
| | SD | 4.8 | 4.9 | 4.9 | 5.1 | |

pMCAO, permanent middle cerebral artery occlusion; DW, distilled water; ASA, Aspirin®; A, Ethanol extract of *Coptidis* Rhizoma, *Phellodendri* Cortex, *Scutellariae* Radix, *Gardeniae* Fructus, and *Rhei* Rhizoma; B(K1), Ethanol extract of 2019 Yeongyang *Salviae Mitiorrhizae* Radix and *Notoginseng* Radix; SD, standard deviation. * $p < 0.05$ vs. DW; ** $p < 0.01$ vs. A60 by analysis of variance followed by Tukey's test.

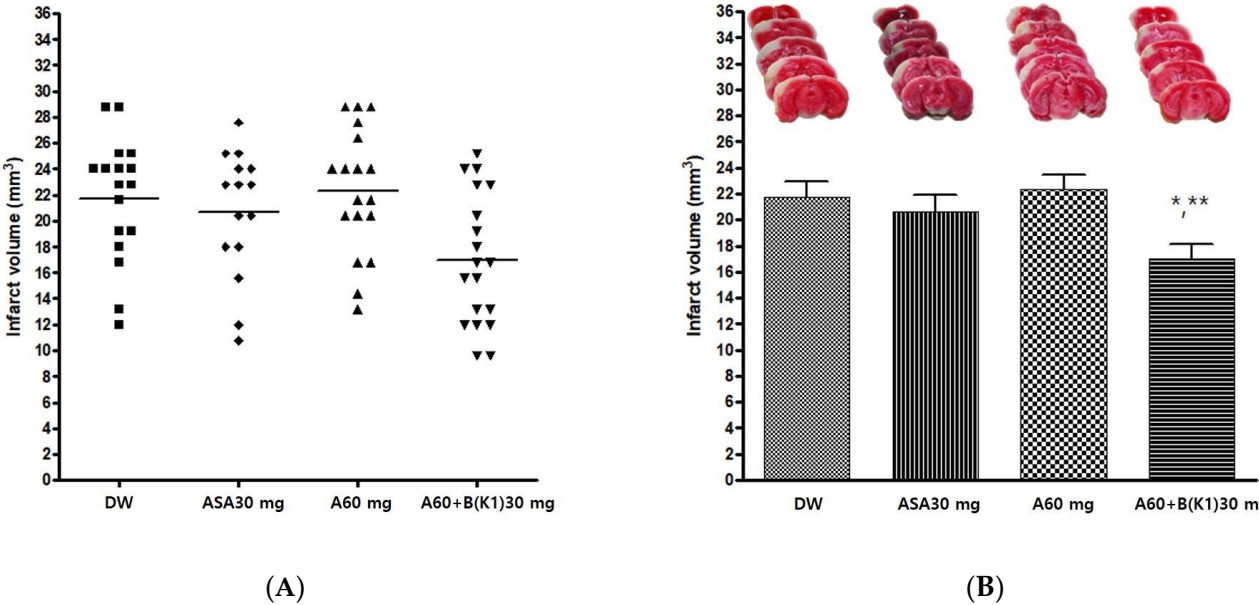

**(A)** **(B)**

**Figure 4.** Effects of experimental drug A60+B(K1) 30 mg/kg on ischemic brain injury after permanent middle cerebral artery occlusion (pMCAO). Infarct volume was measured 24 h after pMCAO. DW and all drugs were administered just before pMCAO. (**A**) Individual infarct volume values for each group (n = 15–19 mice for each group). (**B Upper**) Representative photographs of infarcted brain slices from DW- and drug-treated mice. (**B Lower**) The group of experimental drug A+BK1 exhibited a significant decrease in infarct volume after pMCAO compared to groups DW and A60. DW, distilled water; ASA, Aspirin®; A, Ethanol extract of *Coptidis* Rhizoma, *Phellodendri* Cortex, *Scutellariae* Radix, *Gardeniae* Fructus, and *Rhei* Rhizoma; B(K1), Ethanol extract of 2019 Yeongyang Korean *Salviae miltiorrhizae* Radix and *Notoginseng* Radix. Data represent means ± standard deviation. * $p < 0.05$ vs. DW; ** $p < 0.01$ vs. A60 by analysis of variance followed by Tukey's test.

3.2.4. Effects of Experimental Drug A+BK2 on Infarct Volume after pMCAO

The experimental drug A 60 mg/kg and BK2 30 mg/kg combination (A60+B(K2)30) was administered orally before pMCAO surgery to confirm the neuroprotective effect. Distilled water (DW), Aspirin® 30 mg/kg (ASA30), and drug A 60 mg/kg (A60) were also administered in the same manner as controls. The infarct volume was measured 24 h after pMCAO. The group administered the experimental drug A+BK2 showed a significant decrease in infarct volume after pMCAO compared with other groups ($p = 0.0072$). Post-hoc analysis revealed that the group administered the experimental drug A+BK2 exhibited a significant decrease in infarct volume after pMCAO compared to the groups administered DW and A60 ($p < 0.05$ and $p < 0.01$, respectively) (Table 6 and Figure 5).

**Table 6.** Effects of experimental drug A60+B(K2) 30 mg/kg on infarct volume after pMCAO.

| Variables | | Experimental Drugs (mg/kg) | | | | *p*-Value |
|---|---|---|---|---|---|---|
| | | DW (n = 17) | ASA30 (n = 15) | A60 (n = 18) | A60+B(K2)30 (n = 19) | |
| Infarct volume | Mean (mm³) | 21.7 | 20.6 | 22.3 | 17.4 *, ** | 0.0072 |
| | SD | 4.8 | 4.9 | 4.9 | 3.7 | |

pMCAO, permanent middle cerebral artery occlusion; DW, distilled water; ASA, Aspirin®; A, Ethanol extract of *Coptidis* Rhizoma, *Phellodendri* Cortex, *Scutellariae* Radix, *Gardeniae* Fructus, and *Rhei* Rhizoma; B(K2), Ethanol extract of 2019 Jangheung *Salviae Mitiorrhizae* Radix and *Notoginseng* Radix; SD, standard deviation. * $p < 0.05$ vs. DW; ** $p < 0.01$ vs. A60 by analysis of variance followed by Tukey's test.

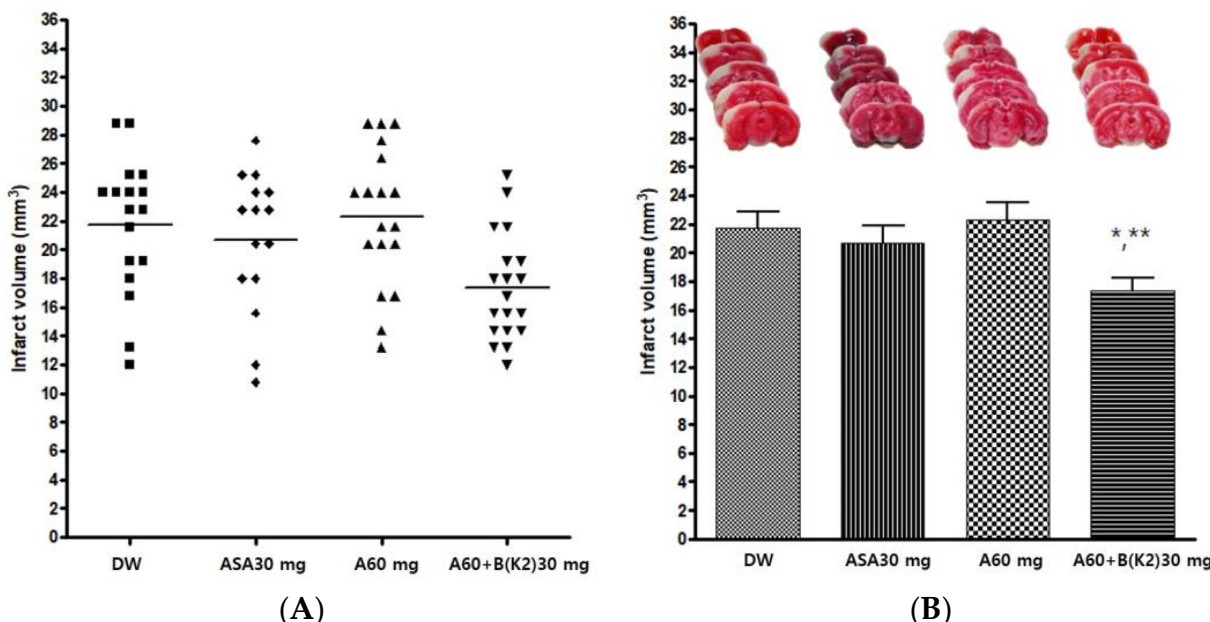

**Figure 5.** Effects of experimental drug A60+B(K2) 30 mg/kg on ischemic brain injury after permanent middle cerebral artery occlusion (pMCAO). Infarct volume was measured 24 h after pMCAO. DW and all drugs were administered just before pMCAO. (**A**) Individual infarct volume values for each group (n = 15–19 mice for each group). (**B Upper**) Representative photographs of infarcted brain slices from DW- and drug-treated mice. (**B Lower**) The group of experimental drug A+BK2 exhibited a significant decrease in infarct volume after pMCAO compared to groups DW and A60. DW, distilled water; ASA, Aspirin®; A, Ethanol extract of *Coptidis* Rhizoma, *Phellodendri* Cortex, *Scutellariae* Radix, *Gardeniae* Fructus, and *Rhei* Rhizoma; B(K2), Ethanol extract of 2019 Jangheung Korean *Salviae miltiorrhizae* Radix and *Notoginseng* Radix. Data represent means ± standard deviation. * $p < 0.05$ vs. DW; ** $p < 0.01$ vs. A60 by analysis of variance followed by Tukey's test.

### 3.2.5. Effects of Experimental Drug A+BK3 on Infarct Volume after pMCAO

The experimental drug A 60 mg/kg and BK3 30 mg/kg combination (A60+B(K3)30) was administered orally before pMCAO surgery, to confirm the neuroprotective effect. Distilled water (DW), Aspirin® 30 mg/kg (ASA30), and drug A 60 mg/kg (A60) were also administered in the same manner as controls. The infarct volume was measured 24 h after pMCAO. The group administered the experimental drug A+BK3 showed a significant decrease in infarct volume after pMCAO compared with other groups ($p = 0.0287$). Post-hoc analysis revealed that the group administered the experimental drug A+BK3 exhibited a significant decrease in infarct volume after pMCAO compared to the A60 group ($p < 0.05$) (Table 7 and Figure 6).

**Table 7.** Effects of experimental drug A60+B(K3) 30 mg/kg on infarct volume after pMCAO.

| Variables | | Experimental Drugs (mg/kg) | | | | *p*-Value |
|---|---|---|---|---|---|---|
| | | DW (n = 17) | ASA30 (n = 15) | A60 (n = 18) | A60+B(K3)30 (n = 11) | |
| Infarct volume | Mean (mm³) | 21.7 | 20.6 | 22.3 | 16.7 * | 0.0287 |
| | SD | 4.8 | 4.9 | 4.9 | 5.7 | |

pMCAO, permanent middle cerebral artery occlusion; DW, distilled water; ASA, Aspirin®; A, Ethanol extract of *Coptidis* Rhizoma, *Phellodendri* Cortex, *Scutellariae* Radix, *Gardeniae* Fructus, and *Rhei* Rhizoma; B(K3), Ethanol extract of 2020 *Salviae Mitiorrhizae* Radix and *Notoginseng* Radix; SD, standard deviation; * $p < 0.05$ vs. A60 by analysis of variance followed by Tukey's test.

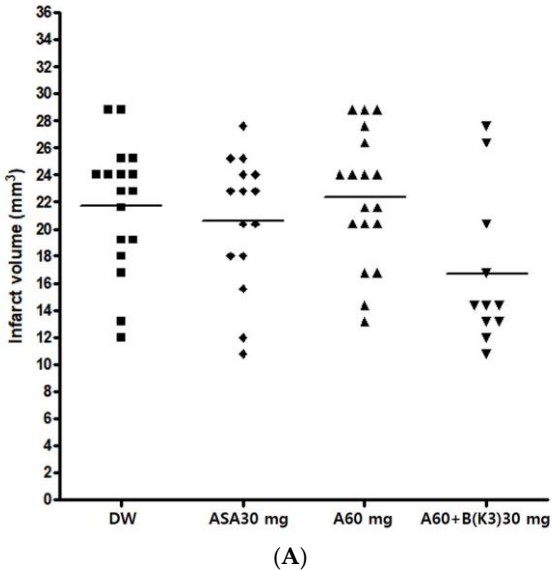
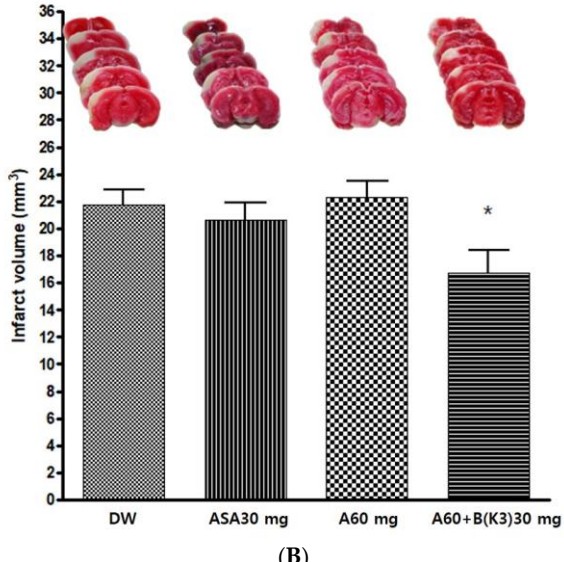

(**A**)            (**B**)

**Figure 6.** Effects of experimental drug A60+B(K3) 30 mg/kg on ischemic brain injury after permanent middle cerebral artery occlusion (pMCAO). Infarct volume was measured 24 h after pMCAO. DW and all drugs were administered just before pMCAO. (**A**) Individual infarct volume values for each group (n = 15–19 mice for each group). (**B Upper**) Representative photographs of infarcted brain slices from DW- and drug-treated mice. (**B Lower**) The group administered the experimental drug A+BK3 exhibited a significant decrease in infarct volume after pMCAO compared to the A60 group. DW, distilled water; ASA, Aspirin®; A, Ethanol extract of *Coptidis* Rhizoma, *Phellodendri* Cortex, *Scutellariae* Radix, *Gardeniae* Fructus, and *Rhei* Rhizoma; B(K3), Ethanol extract of 2020 Korean *Salviae miltiorrhizae* Radix and *Notoginseng* Radix. Data represent means ± standard deviation. * $p < 0.05$ vs. A60 by analysis of variance followed by Tukey's test.

## 4. Discussion

In this study, the pMCAO model was used to compare differences in the salvianolic acid B content of *Salviae miltiorrhizae* Radix based on the cultivation region and the neuroprotective effect of GCD based on salvianolic acid B content. The salvianolic acid B content in *Salviae miltiorrhizae* Radix varied among different regions. GCD with high salvianolic acid B showed significant neuroprotective effects after pMCAO, whereas GCD with low salvianolic acid B did not.

In the analysis of salvianolic acid B, *Salviae miltiorrhizae* Radix C1 and C2 had 2.7% and 2.4% of salvianolic acid B, respectively, and K1, K2, and K3 included 7.4%, 6.2%, and 7.5%, respectively. The Korean Pharmacopeia defines the content of salvianolic acid B in *Salviae miltiorrhizae* Radix as >4.1% [36] and the Chinese Pharmacopeia defines it as over 3.0% [37]. C1 and C2 did not meet the Korean or Chinese Pharmacopeia standards, whereas K1, K2, and K3 did. Yang et al.'s [35] comparison of the contents of salvianolic acid B in three types of *Salviae miltiorrhizae* Radix from Korea and one from China revealed that samples from Korea ranged from 4.4% to 6.3%, and those from China were 2.6%, which are similar to our results. However, Hu et al. [41] reported that salvianolic acid B analysis of seven *Salviae miltiorrhizae* Radix varieties from different cultivation regions in China ranged from 4.4 to 6.7%. Seong et al. [42] compared five *Salviae miltiorrhizae* Radix varieties from Korea and three from China and reported that the average content of salvianolic acid B from Korea was 5.53% and that from China was 4.42%. Thereby, the quality of *Salviae miltiorrhizae* Radix may have varied greatly, depending on the cultivation region.

Comparison of the neuroprotective effect of each experimental drug revealed that pretreatment with GCD with high salvianolic acid B provided significant neuroprotective effects after pMCAO, whereas pretreatment of GCD with low salvianolic acid B did not. The neuroprotective effect of GCD pretreatment was proportional to the salvianolic acid B content. Salvianolic acid B has previously been proven to have an ischemic/reperfusion

neuroprotective effect in a middle cerebral artery occlusion (MCAO) model, similar to that in the present study [43]. This supports the hypothesis of this study that the difference in salvianolic acid B content led to the difference in neuroprotective effects. Therefore, it can be concluded that there is a need to secure high-quality *Salviae miltiorrhizae* Radix raw materials for effective and stable quality control and stability of the neuroprotective effects of GCD. Additionally, in previous studies, differences in salvianolic acid B content were reported to occur according to the extraction solvent of *Salviae miltiorrhizae* Radix [35] and to depend on the drying conditions of *Salviae miltiorrhizae Radix* [44]. Therefore, to standardize and stabilize GCD, it is necessary to standardize not only the marker compound content, but also the extraction and drying methods of *Salviae miltiorrhizae* Radix. Further research on these aspects is needed.

In contrast, similar to previous studies, neuroprotective effects did not differ between the A60 and the control groups [22,23]. Meanwhile, GCD containing BK1 and BK2 had a significant neuroprotective effect after pMCAO compared to aspirin used as a control group. In this study, aspirin as a control did not have a significant neuroprotective effect compared to distilled water, similar to a previous study [23]. There is much controversy regarding the primary preventive effects of aspirin, which is used to prevent cerebral infarction [45–47]. Therefore, for the primary prevention of cerebral infarction, it is suggested that the role of GCD using high-quality *Salviae miltiorrhizae* Radix may be greater than that of aspirin. Additional complimentary research on this point is needed.

Although the mechanism of GCD's neuroprotective effect has not been elucidated, a previous study suggested an anti-inflammatory mechanism through inhibition of microganglia activation and improvement of collateral blood flow [22]. Various mechanisms have been revealed regarding the neuroprotective effect of salvianolic acid B. Oxidative stress due to secretion of reactive oxygen species (ROS) is a major destructive factor for neural cell apoptosis induced by ischemic brain injury [48]. A study using the pMCAO model demonstrated that salvianolic acid B protects brain tissue in acute ischemia with antioxidant action through reduction of ROS [49]. In addition, since neuroinflammation induced by ischemic stroke initiates a complex series of inflammatory events to induce secondary brain damage, suppression of early inflammatory responses plays an important therapeutic role in ischemic stroke [50]. Pro-inflammatory cytokines, such as tumor necrosis factor-$\alpha$, interleukin (IL)-1$\beta$, and IL-6, are suppressed by salvianolic acid B, which causes anti-inflammatory activity, resulting in neuroprotective effects in ischemic brain damage [49,51]. Additionally, platelet adhesion, activation, and aggregation are critical in arterial thrombosis and, thus, are important in the pathophysiology of ischemic stroke [52]. Salvianolic acid B is known to inhibit platelet activation by reducing P-selectin involved in platelet aggregation [53]. Taken together, this evidence leads to the conclusion that GCD containing Salviae Miltiorrhizae Radix with a high salvianolic acid B content further activates these mechanisms to achieve a better neuroprotective effect.

There are several significant aspects of this study. First, it laid the foundation for homogenizing the efficacy of the GCD. To date, there have been many studies dealing with differences in quality according to the marker compound of medicinal materials, but studies dealing with differences in medicinal effects according to the marker compound are rare. Second, these findings revealed that even with the same drug, drug efficacy may vary depending on which medicinal material was used to manufacture the drug that plays a major role, and thus, effective and stable quality control for GCD can be achieved by securing stable materials. Collectively, all of these processes can help the homogenization, stability, and standardization of GCD, and ultimately contribute to the prevention and treatment of cerebral infarction.

Despite the importance of our findings, future studies concerning the pharmacokinetics of GCD should be conducted to evaluate GCD brain bioavailability and relate to the neuroprotective effects observed. In the current study, it was assumed that drugs administered orally to mice were immediately absorbed and reached the brain to have neuroprotective effects; however, pharmacokinetic experiments on this process were not

performed. Since no previous study has revealed the pharmacokinetics of GCD, the basis for assumptions regarding the pharmacokinetics of GCD should be further elucidated, experimentally. In addition, since the evaluation of the neuroprotective effect was performed by measuring only infarct volume after pMCAO, research to verify the effect by other methods is needed to supplement the results of the current study. Although infarct volume is an important indicator in evaluating the effectiveness of a drug for the treatment of stroke, the results may be less convincing with only one evaluation index. Finally, because an in vivo model was used in this study, additional verification of the evidence presented in this study is necessary through future clinical studies.

## 5. Conclusions

This study analyzed the salvianolic acid B content of five samples of *Salviae miltiorrhizae* Radix and compared the difference in neuroprotective effects of GCD according to the salvianolic acid B content using the pMCAO model. The content of salvianolic acid B was high in three samples of *Salviae miltiorrhizae* Radix (K1, K2, and K3) and met the Korean and Chinese standards, whereas those in samples C1 and C2 did not. GCDs with low salvianolic acid B (BC1 and BC2) showed no significant change in neuroprotective effects compared to that of the control groups. However, GCDs with high salvianolic acid (BK1, BK2, and BK3) showed marked superior neuroprotective effects. These findings indicate that securing a stable raw material of *Salviae miltiorrhizae* Radix is important for the homogenization of GCD.

**Author Contributions:** Conceptualization, S.K. and H.-G.L.; methodology, S.K., S.-Y.C., J.-M.P. and C.-N.K.; validation, S.K. and H.-G.L.; formal analysis, S.K. and H.-G.L.; investigation, S.K., W.-S.J., S.-K.M. and K.-H.C.; data curation, S.K. and H.-G.L.; writing—original draft preparation, S.K. and H.-G.L.; writing—review and editing, H.-G.L., S.K., S.-U.P., W.-S.J., S.-K.M. and K.-H.C.; supervision, S.K.; project administration, S.K.; funding acquisition, S.K. All authors have read and agreed to the published version of the manuscript.

**Funding:** This research was supported by a grant from the Korea Health Technology R&D Project through the Korea Health Industry Development Institute (KHIDI), funded by the Ministry of Health and Welfare, Republic of Korea (No. HF22C0076).

**Institutional Review Board Statement:** The animal study protocol was approved by the Kyung Hee Medical Center Institutional Animal Care and Use Committee (KHMC-IACUC 19-004, approved on 28 February 2019).

**Data Availability Statement:** The data presented in this study are available on request from the corresponding author. The data are not publicly available for the protection of personal information.

**Acknowledgments:** This paper is based on Han-Gyul Lee's Ph.D. dissertation.

**Conflicts of Interest:** The authors declare no conflict of interest. The funders had no role in the design of the study; in the collection, analyses, or interpretation of data; in the writing of the manuscript; or in the decision to publish the results.

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
