# Peer review of "Neuroprotective Effects of Geopung-Chunghyuldan Based on Its Salvianolic Acid B Content Using an In Vivo Stroke Model"

_cimb, doi:10.3390/cimb45020104_

Round 1
Reviewer 1 Report
The submitted article ‘Neuroprotective Effects of Geopung-Chunghyuldan Based on Its Salvianolic Acid B Content Using an In Vivo Stroke Model’ is interesting and good article. This is a kind of rare studies, hence is very desire. This study found that even with the same drug, differences in drug efficacy may occur depending on which medicinal material was used for the drug that plays a major role, and thus, effective and stable quality control for GCD can be achieved by securing stable materials. Collectively, all of these processes can help the homogenization, stability, and standardization of GCD and ultimately contribute to the prevention and treatment of cerebral infarction. An in vivo model was used in this study, hence additional verification of the evidence presented in this study is necessary through clinical studies in the future. It will be appropriate for Current Issues in Molecular Biology (MDPI). Below I pointed most of mistakes and matters for explanation.
1. In introduction the strong emphasis should be placed on justification
2. The good idea should be summary as graphical workflow in introduction because this topic is complex, perhaps chart/graph will be good idea
3. Figure 1 should be more readable - perhaps zoom should be applied for critical area?
4. Conclusion part should be more informative (more details, perhaps in brackets?)
5. Please include advances and disadvantages of your studies in conclusions
I totally agree that this is very important subject, and it is very important for publication in MDPI. I will recommend this article for publication in Current Issues in Molecular Biology (MDPI) after minor revision.
Reviewer 2 Report
This manuscript has potential, however, there is need for serious improvement prior to reconsideration for publication. The authors should pay attention to the following points.
1. How long was the experimental drug administered before the pMCAO surgery?
2. No explanation is provided on the mechanism behind the neuroprotective effect of the extract.
3. Please it is imperative that the objectives and conclusions of this work be re-written clearly.
4. The entire manuscript should be rigorously edited by a professional for English language and grammatical errors.
Reviewer 3 Report
Comments and Suggestions for Authors
General comment:
Lee and colleagues report the neuroprotective effects of Geopung-Chunghyuldan due to its content in salvianolic acid B in an animal model of stroke. After analysing the content of salvianolic acid B of five kinds of Salviae Miltiorrhizae Radix, they found that only three of these showed to comply with Korean and Chinese Pharmacopeia. Then, the authors assessed the neuroprotective effects of the five samples (drug B) with different salvianolic acid B content by measuring the infarct volume after pMCAO surgery. Formulations with higher concentration of salvianolic acid B reduced the infarct volume.
Overall, the authors’ findings show that salvianolic acid B has neuroprotective effects in stroke. Also, this work highlights the great variation of bioactive compounds in their sources and the importance of having standards.
Major issues:
1. This review needs some English proofreading.
Minor issues:
1. In Abstract section: substitute “the present study aims” to “the present study aimed”.
2. In Abstract section: Conclusions lack something about the relevance of the observations performed in the work regarding salvianolic acid B in stroke. Rephrase or add a sentence.
3. In Introduction section (1st line): substitute “Cerebral infarction” by “Stroke”.
4. In Introduction section (last paragraph): add “in five different types of Salviae Miltiorrhizae Radix” after “the salvianolic acid B content was analyzed”.
5. In Discussion section (last paragraph): Please rephrase this last paragraph. For example, replace “This study has a number of limitations” by “Despite the importance of our findings, future studies concerning the pharmacokinetics of GCD should be conducted to evaluate GCD brain bioavailability and relate to the neuroprotective effects observed.” Use the same logic in the next sentences ...
6. Revise the manuscript for grammatical errors.
Round 2
Reviewer 2 Report
Accept.